# Curcumin-Based Nanoformulations: A Promising Adjuvant towards Cancer Treatment

**DOI:** 10.3390/molecules27165236

**Published:** 2022-08-16

**Authors:** Salar Hafez Ghoran, Andrea Calcaterra, Milad Abbasi, Fatemeh Taktaz, Kay Nieselt, Esmaeil Babaei

**Affiliations:** 1Phytochemistry Research Center, Shahid Beheshti University of Medical Sciences, Tehran 16666-63111, Iran; 2Medicinal Plant Breeding and Development Research Institute, University of Kurdistan, Sanandaj 66177-15175, Iran; 3Department of Chemistry and Technology of Drugs, Sapienza–University of Rome, P. le Aldo Moro 5, 00185 Rome, Italy; 4Department of Medical Nanotechnology, School of Advanced Medical Sciences and Technologies, Shiraz University of Medical Sciences, Shiraz 71336-54361, Iran; 5Department of Biology, Faculty of Sciences, University of Hakim Sabzevari, Sabzevar 96179-76487, Iran; 6Department of Advanced Medical and Surgical Sciences, University of Campania “Luigi Vanvitelli”, 80138 Naples, Italy; 7Interfaculty Institute for Bioinformatics and Medical Informatics (IBMI), University of Tübingen, 72076 Tübingen, Germany; 8Department of Biology, Faculty of Natural Science, University of Tabriz, Tabriz 51666-16471, Iran

**Keywords:** nanocurcumin, cancer treatment, nanocarriers, drug delivery system, curcumin nanoformulations, cancer immunotherapy

## Abstract

Throughout the United States, cancer remains the second leading cause of death. Traditional treatments induce significant medical toxic effects and unpleasant adverse reactions, making them inappropriate for long-term use. Consequently, anticancer-drug resistance and relapse are frequent in certain situations. Thus, there is an urgent necessity to find effective antitumor medications that are specific and have few adverse consequences. Curcumin is a polyphenol derivative found in the turmeric plant (*Curcuma longa* L.), and provides chemopreventive, antitumor, chemo-, and radio-sensitizing properties. In this paper, we summarize the new nano-based formulations of polyphenolic curcumin because of the growing interest in its application against cancers and tumors. According to recent studies, the use of nanoparticles can overcome the hydrophobic nature of curcumin, as well as improving its stability and cellular bioavailability in vitro and in vivo. Several strategies for nanocurcumin production have been developed, each with its own set of advantages and unique features. Because the majority of the curcumin-based nanoformulation evidence is still in the conceptual stage, there are still numerous issues impeding the provision of nanocurcumin as a possible therapeutic option. To support the science, further work is necessary to develop curcumin as a viable anti-cancer adjuvant. In this review, we cover the various curcumin nanoformulations and nanocurcumin implications for therapeutic uses for cancer, as well as the current state of clinical studies and patents. We further address the knowledge gaps and future research orientations required to develop curcumin as a feasible treatment candidate.

## 1. Introduction

In the United States, there were 1,806,590 new cancer diagnoses and 606,520 cancer deaths in 2020. Cancer mortality rates rose until 1991 and then fell steadily until 2017, resulting in a 29% drop in overall deaths. That means there were 2.9 million fewer cancer deaths than there would have been if high incidence had continued [1]. Traditional clinical procedures for treating or removing tumors, such as chemotherapy, radiotherapy, and surgical dissection, are widely used [2]. Although chemotherapy remains a very efficient strategy in the battle against cancer, it is typically associated with substantial drawbacks and side effects. Therefore, there remains the possibility that tumors that have been surgically removed may return and may also be resistant to radiation treatment [3]. Marine and terrestrial natural products are far more favorable for improving treatment outcomes in disorders with complicated pathological behavior. Due to their biochemical structures, natural compounds can affect numerous cellular targets, such as genes and proteins [4,5]. Herbal preparations and their related natural products offer a pivotal role in cancer chemotherapy and chemoprevention. As they are safe and able to minimize the side effects on healthy cells, selected medicinal plants have been developed to address a variety of malignancies [6,7]. Most significantly, the development of novel therapeutic approaches is critical for accurately treating tumors and preventing cancer from progressing to the metastatic level. Researchers are investigating herbal medicines, natural medicine, and nutrition to address the typical main difficulties in traditional cancer therapy [8,9]. Curcumin is the primary polyphenol isolated from the rhizomes of *Curcuma longa* L. (family; Leguminosae), which has a long-standing tradition of usage throughout East Asian nations as a curry powder (turmeric) [10]. Curcumin has a variety of therapeutic applications, exhibiting anti-inflammatory, antioxidant, antidiabetic, antibacterial, antifungal, antiproliferative, anticancer, and hepatoprotective effects [11]. Curcumin has demonstrated considerable cancer-suppressive efficacy in vitro and in vivo against various cancer growths at all stages, including initiation, promotion, and propagation [12,13,14]. Curcumin’s hydrophobicity makes it difficult for it to cross the cellular membranes, because it bonds to the membrane lipid fatty acyl chains via hydrophobic interactions and hydrogen bonding. Therefore, the levels of curcumin throughout the cytoplasm remain very low. Curcumin nanosystems improve the nature of curcumin as a therapeutic agent, overcoming such challenges while increasing bioavailability. Interestingly, the entrapment and nanodrug-loading efficacy are highly reliant on the manufacturing process and the carrier structure used to carry the nano drugs. Both are critical in drug distribution and have a significant effect on the volume and magnitude of the release profile from the transporter. As loading capacity is correlated with the drug-to-carrier system proportions, encapsulation efficiency indicates how most of the medication inside the nanostructures is easily adsorbed or entrapped [15,16]. Several investigations have shown the viability of employing nanotherapeutic methods to enhance in vivo and in vitro curcumin application using polymers, liposomes, cyclodextrins, conjugates, dendrimers, micelles, and nanostructures [17,18,19]. Since 2011, most curcumin nanoformulations have been studied and are sometimes used in clinical trials [20]. Researchers initially concentrated on improving the absorption rate constant (K_a_) of curcumin, but subsequently turned their attention to the successful targeting of curcumin to the diseased region using an antibody, aptamer, or peptide mediation assistance. The oral bioavailability of curcumin encapsulated within poly(lactic-co-glycolic acid) nanomaterials was investigated, and nanocurcumin’s therapeutic efficacy was found to be nine times more potent than free curcumin [21]. Experiments further show that nanocurcumin is efficient for liver diseases [22], as well as for brain tumors [23], cardiac diseases [24], and cancers [25].

In this review, we performed a comprehensive literature search for the latest information on the curcumin nanoformulations and the abovementioned keywords using the following databases: Scopus, Web of Science, PubMed, and Google Scholar. The extra searched keywords were “cancer immunotherapy”, “anti-angiogenesis”, “clinical trials”, “curcumin patent”, and “mechanism of action”. Aside from briefly highlighting the molecular targets of curcumin and strategies for curcumin nanoformulation synthesis, we aim to summarize and discuss numerous curcumin nanoformulations and comparable properties of the nanocurcumin and curcumin, which reveal the different therapeutic effects of nanocurcumin. In the final sections, we examine the currently ongoing clinical investigations, the scientific disparity, and probable future research necessary to endorse curcumin as a promising therapeutic adjuvant.

## 2. Curcumin as Hydrophobic Phytochemical in Medicine

### 2.1. Curcumin Structure 

The chemical structure of curcumin, known as (1*E*,6*E*)-1,7-bis(4-hydroxy-methoxyphenyl)-1,6-heptadiene-3,5-dione, consists of two aromatic ring structures containing hydroxy and methoxy groups, which are linked via a seven-carbon-containing chain of a moiety of α,β-unsaturated β-diketone (Figure 1) [26,27]. In general, commercially available curcumin is a combination of three curcuminoids—bisdemethoxycurcumin (around 5%), demethoxycurcumin (around 18%), and diferuloylmethane (around 77%) [28]. Curcumin shows keto–enol tautomerism, with the keto form predominating in neutral or acidic environments, and the consistent enol type predominating in an alkaline environment [29]. Curcumin is less water-soluble at neutral or acidic pH due to its ionization properties (pK_a_ = 7.8; 8.5; 9.0) [30], but it dissolves in alkali ethanol, methanol, ketone, chloroform, dimethyl sulfoxide, acetone, and acetic acid [31]. Due to multiple methoxy replacements within the diferuloylmethane chemical composition, which are responsible for the yellow color of curcumin, the curcuminoids bisdemethoxycurcumin, de-methoxycurcumin, and cyclocurcumin are pivotal for the specific pharmacological and biological distinguishable behavioral patterns of these substances [32]. According to Somparn et al., the antioxidant activity of the curcumin analogues is in the following descending order: diferuloylmethane > demethoxycurcumin > bisdemethoxycurcumin [33]. Transition metal chelation, which is bound to the *O*-methoxy phenols and diketone moieties, is a well-known antioxidant pathway found in curcumin. Bisdemethoxycurcumin, demethoxycurcumin, and diferuloylmethane inhibit nuclear factor-kappa B (NF-κB) and hemeoxygenase-1. They are also used as Michael acceptors in α,β-conjugated addition reactions [34,35]. 

### 2.2. Molecular Targets of Curcumin 

In addition to the Food and Drug Administration (FDA), many research papers go into great depth regarding the inherently well-tolerated, safe, and biocompatible characteristics of pure curcumin [36,37,38]. So far, both pre-clinical and clinical studies of curcumin administration via the oral route have demonstrated the poor biodistribution of curcumin [39,40]. A typical example is a pharmacokinetic investigation performed on healthy human subjects, which revealed that even after high oral-dosage administration (C_max_) of 10 g and 12 g of curcumin, only 2.30 and 1.73 μg/mL of curcumin were found in the blood. This means curcumin undergoes several physiological changes as it passes through the liver and gut [41]. Curcumin also affects a range of important targets, including protein kinase C (PKC), thioredoxin reductase, tubulin, 5-lipoxygenase, COX-2, cytokines, transcription factors, enzymes, growth factors and their receptors, and genes involved in cellular proliferation and apoptosis [42,43,44].

To extend the topic, molecular mechanisms underlying curcumin’s effects on the upregulation of pro-apoptotic proteins include p53 and Bax. Curcumin may induce apoptosis by activating p53 and downregulating PI3K, p-Akt, and p-mTOR substrates, in which PI3k/Akt/mTOR plays a salient role in cancer cell survival pathways [45]. Curcumin directly activates p53, which repairs DNA, and inhibits the protein kinase B (Akt), which leads to increased expression of Bax genes and apoptosis induction [46]. Furthermore, NF-κB is a pro-inflammatory transcription factor that modulates the expression of IL-1 and IL-2 and interferon-y, which participate in many cell signaling pathways, such as cancer progression and inflammation. Curcumin downregulates the activator protein-1, which is associated with anti-apoptotic, mitogenic, and pro-angiogenic genes. Curcumin also suppresses NF-κB activity by inhibiting NF-κB kinase activation and blocking the p65 subunit of NF-κB [47]. Curcumin also modulates several pathways hyperactivated in cancer stem cells, such as the Sonic Hedgehog, Wnt/β-catenin, Notch-1, AR, and RTK pathways [48,49]. Matrix metalloproteinases (MMPs), as members of the zinc-dependent endopeptidases, are other curcumin targets that are over-expressed in tumor infiltration. Most importantly, MMP2 and MMP9 are involved in tumor angiogenesis when the extracellular matrix is degraded. In human breast cancer epithelial cells, regardless of suppression of MMPs expression, curcumin inhibits the TPA-induced activation of ERK and NF-κB transcriptional activation. Furthermore, curcumin downregulates MMP9 expression by inhibiting NF-κB and AP-1 binding to the DNA promoter domain in brain tumors [50]. Moreover, curcumin exerts anticancer properties by affecting cyclin D1, which can act as a transcriptional co-regulator and regulates cell cycle progression. It has been shown that high levels of cyclin D1 are associated with cancer development and progression. Indeed, inhibition of cyclin D1 by curcumin occurs through NF-κB suppression [51]. Interestingly, the anti-proliferation effects of curcumin have been documented in multiple studies as modulating several molecules, such as STAT-3, Forkhead Box O3 (FOXO3a), EGFR, and Eukaryotic initiation Factor (eIFs), and transforming growth Factor-Beta (TGF-β) [52]. For example, curcumin treatment suppresses the STAT3 phosphorylation in SCLCNCI-H446 and NCI-1688 cancer cells and decreases cyclin B1, which promotes cell cycle progression from G2 to M phases and then inhibits cell proliferation [53]. Several studies reported that curcumin upregulated the expression of FOXO3a in A549 and H460 human lung cancer cells by increasing ROS production. On the other hand, high levels of FOXO3a enhanced target genes, such as p27, Bim, and p2 [54]. As a crucial signaling pathway, JAK2/STAT3 plays a critical role in various stages of cancer development. As a result of inhibiting the JAK/STAT3 signaling pathway, curcumin inhibited JAK2 activity and reduced tumorspheres [55]. An increase in superoxide dismutase and a decrease in malondialdehyde and 4-hydroxynonenal were also observed after curcumin treatment of human A549 lung adenocarcinoma cells, and were shown to induce apoptosis. Moreover, in response to curcumin, JNK and p38 phosphorylation increases, while ERK phosphorylation decreases. The results showed that curcumin induced the activation of p38, JNK, ERK, and MAPK pathways, which were involved in the induction of apoptosis of A549 cells [56]. Recent evidence also suggests that curcumin’s antitumor properties are mediated by miRNAs. For instance, miRNA-194-5p is downregulated in gastric cancer, while the miRNA-194-5p over-expression suppresses tumor growth [57]. Li et al. demonstrated that curcumin suppresses the progression of gastric cancer by regulating circ-0056618. In curcumin-treated gastric cells, over-expression of circ-0056618 promoted cell proliferation, migration, and invasion and suppressed apoptosis and cell cycle arrest. However, miR-194-5p as the target for circ-0056618 increases after curcumin treatment, so curcumin treatment delayed the development of gastric cells by downregulating circ-oo56618 and upregulating miR-194-5p [58]. A summary of molecular targets of curcumin is shown in Figure 2.

## 3. Advances in Curcumin Formulations

### 3.1. Synthesis of Curcumin Nanomaterials

An array of methodologies has been developed to synthesize nanocurcumin. Fessi method, nanoprecipitation, thin-film hydration, microemulsion, emulsion polymerization, ultra-sonication, wet milling, spray drying, antisolvent precipitation, ionic gelation, solvent evaporation, coacervation process, single emulsion, ionic gelation, and solid dispersion are some of the most popular techniques [59,60,61,62]. According to in-depth literature evidence, curcumin nanostructures demonstrate better stability and solubility. The tendency of polymeric materials to cross-link in the existence of counterions is a basis for the ionic gelation procedure. This method has appeared as one of the most effective methods for producing biocompatible, non-toxic, and environmentally friendly natural polymers (alginate and chitosan) [62,63,64]. Therefore, many experiments have been conducted to examine the ability or utilization of natural-based polymeric nanostructures (alginate/chitosan) for curcumin’s oral delivery [65,66,67,68]. Das et al. designed the curcumin-based nanoformulation of tripolymeric composites (chitosan, alginate, and pluronic) as well as their transmission into cancerous cells utilizing an ionic gelation strategy [68,69]. Curcumin-bound chitosan nanostructures were developed by Akhtar et al., who showed the effectiveness of employing this strategy to increase antimalarial function in mice while also improving metabolic durability and biocompatibility or bioavailability [69]. Antisolvent precipitation has become another commonly cost-effective and viable method for making curcumin nanostructures, and its effectiveness is determined by stirring speed, temperature, and time [70]. It also improves the curcumin nanostructures’ stability and solubility. It can be used in the industrial processing of pharmaceutical nanomaterials because it is an easy-to-use method [71,72].

### 3.2. Nano-Based Formulation Strategies of Curcumin

Previous research found that the hydrophobic nature of nanostructures influenced the durability and bioavailability of nanomedicines. As a result, it is a salient target in terms of the manipulation of drug carriers [73,74]. Many curcumin nanoformulations have been produced over the last few years. Most of them aim to improve curcumin solubility and bioavailability while also protecting it from hydrolysis inactivation. Such formulations are intended for long-term maintenance and circulation throughout the body, whereas others have focused on intracellular release and cellular transportation processes. Multiple curcumin-based nanoformulations have provided a significant effect on medicinal implementations, which are shown to be effective in the detection of many human disorders. They are defined and addressed in the following (Figure 3).

#### 3.2.1. Micelle Structures

A micelle is a spherical vesicle made up of surfactant molecules with an amphiphilic property that assembles spontaneously in the water [75,76]. Commonly, it is chosen to transmit loaded drugs that are not highly soluble in water, such as curcumin. Curcumin-encapsulated polymeric micelles were synthesized using a solid dispersion technique in a one-step process. The efficacy of generated micelles was assayed on breast cancer [77]. They were more effective than unformulated curcumin at inhibiting the development of breast tumors and uncontrolled spontaneous pulmonary metastatic spread. The solid dispersion of micelles comprising curcumin-poly(ethylene glycol) methyl ether improved curcumin’s anti-tumor and anti-angiogenesis effects. Curcumin micelles can be advantageous during the treatment of pulmonary carcinoma [78]. Chang et al. investigated the intracellular localization, cell absorption, and cytotoxic effects of different dimensions of micelles embodied with curcumin in carcinoma cells from the human colon in vitro. The findings showed that smaller micelles loaded with curcumin possess a higher capacity for inducing cytotoxic effects in carcinoma cells of the human colon than bigger micelles. Accordingly, when employing nanoparticles as drug delivery nanosystems, micelle size, drug loading, and release kinetics and cellular uptake are all crucial elements to consider [79]. Curcumin loaded on the ultra-hydrophilic zwitterionic polymeric material poly(sulfobetaine methacrylate) (PSBMA), which conjugated to zein and formed polymeric micelles, had significantly improved durability, cellular absorption, cytotoxicity toward cancerous cells, and pharmacokinetic properties when compared to traditional methods of using curcumin (Figure 3) [80]. Compared to natural curcumin, encapsulated curcumin in monomethoxy poly(ethylene glycol)-poly(3-caprolactone) micelles inhibits CT26 colon carcinoma cell proliferation in vivo [81].

#### 3.2.2. Liposome Structures

Liposomes are globular vesicles with multiple or single phospholipid bilayers covering aqueous systems that strongly mimic the architecture of the cellular membranes. They have suitable delivery mechanisms for bioactive compounds both in vivo and in vitro [82,83]. Increased biodegradability and biocompatibility, low toxic effects, high durability, improved dissolution rate, controlled release/delivery, ability to target individual cells, ease of production, and versatility are only a few of the benefits of liposomes [84,85,86]. As a result, researchers have paid attention to liposomes as a potent drug-carrier mechanism. The liposome’s size varies from 0.025 to 2.5 μm. The amount of medication capsulation within a liposome is determined by the number and size of bilayers and the vesicle diameter, which are significant considerations in determining the liposome’s circulation time [82]. Many experiments have demonstrated that curcumin is solubilized inside the phospholipidic bilayer of the liposome, allowing curcumin to be dispersed throughout the aqueous phase, thereby increasing curcumin’s effectiveness [87,88]. Liposomal formulations have extended plasma circulation to reach the spleen and other organs and tissues to provide a major therapeutic effect and limit non-specific toxicity [89,90]. Extensive research has revealed that liposome-based curcumin was the best platform for treating various cancer diseases. Liposome-based curcumin prevented the development of the MCF-7 breast cancer cell line and the KHOS OS cell line, and had a significant antitumor behavior in both in vivo and in vitro conditions [91]. Tian investigated the anticancer efficacy and biochemical pathways caused by liposomal curcumins in PC-3 human prostate cancer cells [92]. In contrast with free curcumin, the rate of survival of liposomes loaded with curcumin administered to PC-3 cells was poor and time-dependent. Tefas et al. manufactured liposomes co-encapsulating curcumin and doxorubicin (Dox), which inhibited C-26 murine colon cancer cell proliferation and exhibited a better cytotoxic impact than Dox-loaded liposomes [93]. Correspondingly, resveratrol and curcumin liposomes had a high encapsulation performance, a higher polydispersity index (PI), and narrow size distribution [94]. Liposomal nanocarriers for curcumin delivery and photodynamic treatment mediated by blue light-emitting diode were recently combined to achieve ideal bioactivity and anticancer function. Overall, the findings suggested that liposomes could be a safer transporter for curcumin (Figure 3) [95]. Chen et al. generated liposome nanostructures loaded with curcumin and tested them in B16BL6 melanoma cells for antitumor behavior. Liposome nanostructures significantly slowed the growth of B16BL6 melanoma cells. This was primarily attributed to the improved drug distribution facilitated by the intracellular lipid fusion of cellular membranes and particulates. Additionally, it blocked the PI3K/AKT pathway, which is involved in skin cancer [96].

#### 3.2.3. Cyclodextrin Structures

As soluble transporter structures as well as multi-component systems, cyclodextrins (CDs), such as α-, β-, and γ-cyclodextrins, serve to bind drugs non-covalently. They are frequently used to improve pharmaceutical stability and solubility and to administer active medications to cancerous cells. Cyclodextrins are macrocycle-forming oligosaccharides made up of eight (γ-), seven (β-), or six (α-) D-glucopyranose components interconnected by a 1,4-glycosidic bond. Because of their low cost, ease of production, and versatility, γ-CD, β-CD, and their derivatives are commonly employed to deliver pharmaceutical active agents. Many researchers have recently shown the importance of cyclodextrin in the medication delivery systems of curcumin [97,98]. Yallapu et al. designed a β-CD-controlled curcumin medication transmission mechanism and demonstrated that β-CD–curcumin improved the dissemination of curcumin within prostate cancer cells and strengthened its therapeutic outcomes when compared to curcumin alone [17]. Using cell cycle suspension and the pro-apoptotic behaviors of lung cancer cells, Zhang et al. discovered that β-CD–curcumin (CD15) composition showed higher cytotoxic effects than pure curcumin [99]. Furthermore, CD15 may be a platform for improving curcumin distribution and its therapeutic effectiveness for lung cancer. Nanostructures can be synthesized from sulfobutyl-ether-β-cyclodextrin hyaluronic acid and chitosan and employed for the treatment of colorectal and intestinal epithelial cancer cells, either alone or in combination with curcumin. Curcumin nanostructure demonstrates excellent stabilization and encapsulation properties. It also reduces tumor cell proliferation and the cytotoxicity effects of curcumin in human intestinal epithelial cells [100,101,102]. Furthermore, in retinitis pigmentosa, a complex of cyclodextrins with curcumin, having the property of water-solubility, increased the dissolution rate and generated continuous release of the drug. The findings aided in the development of eye drops made from naturally available phytochemicals (Figure 3) [103].

#### 3.2.4. Conjugate Structures

Conjugates are complexes created by linking two or several substances through a covalent bond. Researchers have increased the bioavailability and oral bioaccumulation of curcumin by conjugating curcumin to hydrophilic polymers and small molecules. According to Manju and Sreenivasan, the conjugation of hyaluronic acid with curcumin reduces the effectiveness of gold nanoparticles while increasing the stability and water solubility of the substance [104]. Singh et al. found that esterifying four phenolic hydroxyls increased the bioavailability of curcumin conjugates glycine and piperic acid, and triggered apoptotic cell death in MDA-MB-231 and MCF-7 cell lines by a mitochondrion-mediated mechanism [105]. Muangnoi et al. made a conjugate of glutaric acid with curcumin in the form of a curcumin–glutaric acid prodrug, and evaluated its function on mice. The dissolution rate and anti-nociceptive activity of the prepared prodrug were found to be higher than curcumin alone. A conjugate of gold nanoparticles with polyvinylpyrrolidone and curcumin has recently been shown to inhibit amyloid Aβ (1–6) agglomeration, with increased curcumin bioavailability and loading capacity (80%), as well as greater medication release. This formulation has the potential to be advantageous in the treatment of Alzheimer’s disease (Figure 3) [106].

#### 3.2.5. Nano- and Nanosphere Structures

Nanomaterials, which have a dimension of 1–100 nm and have special biological, chemical, and physical properties, can be used for drug delivery [107,108]. Medications encapsulated within nanostructures have improved pharmacokinetics, dissolution rate, and specific targeting delivery and controlled/guided release [109,110,111]. Other substances, including solid lipids and polymeric, gold, albumin, and magnetic nanostructures, have also been used to improve the therapeutic benefits of curcumin thus far, because they are biodegradable and non-toxic while also possessing a high binding potential and being non-toxic (Figure 3) [109,112,113].

Sun et al. discovered that solid lipid nanostructures based on curcumin showed increased cell absorption, inhibition of malignant cell proliferation, and improved chemical durability and medicine dissolution rate [114]. Lipid curcumin nanomaterials were assessed for antitumor behavior in adenocarcinoma breast cancer cells. In contrast to pure curcumin, lipid curcumin nanomaterials demonstrated potent bioavailability and drug-releasing assistance. Furthermore, these nanocurcumins effectively increased apoptotic cell death of breast adenocarcinoma cells (Figure 3) [115,116].

Polymeric-based nanostructures provide the benefit of being lightweight and highly biocompatible, allowing them to circulate throughout the bloodstream over an extended time. Polyvinyl alcohol (PVA), *N*-isopropylacrylamide (NIPAAM), hydrophobically modified starch, polyethylene glycol monoacrylate [NIPAAM (VP/PEG A)], silk fibroin, poly(lactic-co-glycolic acid) (PLGA), chitosan, and *N*-vinyl-2-pyrrolidone are only a few of the synthetic and natural polymeric materials that have been used to manufacture curcumin nanomaterials. In CAL27 cancer cells, which show resistance to cisplatin, Chang et al. investigated the molecular pathways triggered with PLGA-modified nanoparticles loaded by curcumin [117,118,119,120]. The curcumin-loaded-PLGA nanostructures seemed to regulate reactive oxygen species (ROS) formation as well as the function of multiple drug resistance protein 1 (MDR1) of CAL27 cisplatin-resistant cancerous cells by triggering the endogenous apoptotic route. Compared to pure curcumin, these PLGA-modified curcumin-loaded nanoparticles are more efficient in treating CAL27 cisplatin-resistant cancerous cells, having higher in vitro bioactivity and better in vivo bioavailability. Another study found that polymer-based nanostructures loaded with curcumin designed with cationic-based Eudragit R E100 copolymer had a high binding affinity and cellular absorption, resulting in increased cytotoxic effects. This nanomaterial formula inhibited tumorigenesis and suppressed colon-26 cell growth 19 times more effectively than curcumin alone (Figure 3) [119,121,122,123].

In a xenografted tumor animal study, Kim et al. discovered that human serum albumin nanostructures loaded with curcumin showed better in vivo anticancer efficacy than unformulated curcumin, with no toxic effects [124]. Furthermore, the results of this study indicated that this formulation could be used as a curcumin-based drug delivery nanosystem for cancer therapy. Thadakapally et al. have demonstrated that nanostructures based on PEG-albumin-curcumin have strong anticancer efficacy for breast cancer lines with a steady long-term circulation threshold and improved dissolution rate [125].

Gold nanomaterials have attracted a lot of attention because of their unique catalytic and optical properties, as well as the fact that they are biocompatible and non-toxic [126,127]. In another study, a potential nanocurcumin substance based on gold nanostructures was synthesized by Nambiar et al. in a cell culture medium without or in combination with fetal bovine serum, and its antitumor activity was reported in human prostate cancer cells [128]. The effects of gold nanomaterials containing curcumin on renal cancerous cells were investigated in vitro. These nanocurcumins were effective antitumor candidates, causing apoptotic cell death in the A498 cell line of renal carcinoma [129]. Curcumin-containing gold nanomaterials, obtained by a green-based synthesis, were also assayed on breast cancer (MCF-7) and colon cancer (HCT-116) cell lines. Compared to free curcumin, these nanomaterials showed potent apoptotic and antiproliferative activity against cancer cells [130].

In cancer cells, iron oxide nanomaterial cores coated by curcumin-containing pluronic polymer (F68) and CD demonstrated an increased dissolution rate [131,132,133]. This composition inhibited the mitochondrial membrane’s capacity and generated more ROS than curcumin. It further displayed a significant antitumor effect, as well as magnetic targeting and resonance imaging capabilities. In lymphocyte cells, sustained release of iron oxide nanostructures comprising curcumin and coated by thiolated starch revealed substantial compatibility of the system. Because of its improved medication encapsulation, durability, and maximum loading capacity, it also induced cytotoxic effects in cancer cells [134]. In another study, Fe_3_O_4_-magnetic nanostructures loaded with curcumin displayed considerable solubility and were appropriate for drug release throughout tumoral tissues [135]. Furthermore, this formulation is applied for tumor tissue imaging techniques. Magnetic nanostructures covered by PEGylated curcumin have newly been established as biocompatible anticancer drug delivery systems (Figure 3) [136,137].

Other formulations for enhancing curcumin’s biological activities include yeast cells, nanodisks, nanogels, and complexes of metals [138,139]. Curcumin was determined to be hydrogen-bonded towards the cellular wall when loaded into the cell membrane of *Saccharomyces cerevisiae*. In another study, Paramera et al. investigated the durability of loaded curcumin in yeast cells, discovering that yeast cells protected the curcumin from external stimuli, such as heat, humidity, and light [140,141]. Curcumin nanodisk compounds are an efficient adjuvant for treating mantle cell lymphoma and other cancers [142]. The association of a curcumin nanodisk with glioblastoma multiforme cells, promoted by ApoE primes, resulted in improved curcumin absorption as well as an enhanced biological function [143]. Compared to curcumin alone, the curcumin–nanogel hybrid could destroy cancer cells. Dandekar et al. combined polyvinyl pyrrolidone and hydroxypropyl methylcellulose to create hydrogel nanostructures loaded with curcumin and evaluated their antimalarial behavior on mice [144]. It was discovered that curcumin-loaded hydrogel nanostructures function much better than unformulated curcumin. As compared to pure curcumin, curcumin stacked into nanogels demonstrated better cellular absorption and improved cytotoxicity on MCF7 and huh7 cell lines [145]. As self-assembled capsules of casein nanogels and carboxymethyl cellulose, curcumin is being delivered and manufactured with casein and folic acid using a layer-by-layer procedure to combat skin cancer. The findings revealed improved cytotoxicity, cellular uptake, and apoptotic cell death in melanoma cancer cells (MEL-39) [146]. Palladium (II) complexes produced with curcumin showed a significant anticancer impact on HeLa, MCF-7, and A549 tumoral cells (Figure 3) [147].

### 3.3. Anti-Cancer Activity of Curcumin

Due to the strong therapeutic results of curcumin, as many as 68 clinical studies (as of 3 May 2012) have investigated curcumin focusing on cancer. Curcumin has a relatively minimal toxicity potential in both humans and animals [28]. Like gemcitabine and cisplatin as chemotherapeutic medications, it inhibits the development of malignant cells with IC_50_ ranging from 5 to 30 μM [148,149,150]. A clinical trial involving 15 people with colorectal cancer found that the cancer was unresponsive to curcumin at 3.6 g/day over four months [151]. According to this report, tumor biomarkers and tumorigenesis did not alter. Generally, the experiments found that while curcumin has anti-tumor properties at a rate of 5–30 μM for 1–2 days, owing to curcumin’s better metabolic functions and its poor bioavailability, reaching these levels at the tumor location in human populations has not been possible [152]. Therefore, curcumin delivery must be developed to address these important matters. 

Curcumin nanoformulations have been produced and adapted by different generic and pharmaceutical companies to improve solubility and dissolution rate. To mitigate this problem, cyclodextrin, adjuvant, and some proprietary innovations were originally employed [153,154,155,156]. Piperine, for example, is strongly advised due to its inhibition effect on intestinal and hepatic glucuronidation. Its inclusion resulted in bioavailability increases of 2000% and 154% in humans and rats, respectively [157,158]. There have been several efforts to enhance the bioavailability of curcumin with the aim of increasing its effectiveness, but these approaches may not have the potential to deliver curcumin into tumors. Therefore, curcumin encapsulation within nanostructures is necessary for cancer treatment with potential target-specific moieties [155,159,160]. Curcumin has been shown in several experiments to serve as an inhibiting factor, preventing cancer development in its early stages. It also acts as a suppressive factor, preventing the proliferation of malignant cells during the progression of carcinogenesis. Curcumin’s antitumor pathways are extensive but diverse, involving various phases of the apoptosis processes and cellular growth. Because curcumin has a plethora of targets and functionalities on cellular growth regulatory pathways, it holds a lot of promise as a chemotherapeutic agent against human solid tumors [161,162,163]. Furthermore, curcumin’s activity on a variety of transcription factors, oncogenes, and signaling proteins affects metastatic spread, tumorigenesis, and cancer development at various levels of carcinogenesis, beginning with early repercussions that cause DNA mutation [164,165]. Curcumin slows tumor development by blocking specific signaling pathways, including inhibiting transcription factors involved in tumorigenesis viz. signal transducers, activating protein-1 (AP-1), and transcription (STAT) protein activators [166]. It causes apoptosis by prohibiting misfolded *N*-CoR protein breakdown and disruption of the ubiquitin–proteasome pathway. Protein kinases are another target of curcumin. Curcumin inhibited the mitogen-activated protein kinase function and epidermal growth-factor receptor activity in lung and pancreatic adenocarcinoma cells [167]. Delivery of curcumin and other phytocompounds to the tumor niche is a significant difficulty. Zibaei et al. investigated the tumoricidal potential of Gemini surfactant nanostructures loaded with curcumin in 3D spheroid HT-29 cells. *In vitro*, the findings revealed that Gemini curcumin has the ability to inhibit cell growth and metastatic spread in 3D spheroid HT-29 cells [159]. In another study, the antiproliferative effects of Gemini curcumin and free curcumin were investigated against ovarian cancer (OVCAR-3 cells). The results showed that compared with pure curcumin, Gemini surfactant nanoparticles increase the cellular uptake and effectively suppress the proliferation of OVCAR-3 cells via induction of apoptosis [168]. 

Additionally, Sobhkhizi et al. revealed that dendrosomal nanocurcumin could be used as an anti-tumor medicine in p53-mutant tumor malignancies [169]. Curcumin’s capacity to induce apoptosis, disrupt cell cycle activity, and inhibit proliferation of cancer cells makes it a possible therapy for human breast (especially MCF-7, MDA-MB-231, MDA-MB-468, and SkBr-3 cells), lung, colorectal, prostate, carcinoma, melanoma liver, myeloma, and pancreatic cancers (Figure 4) [170,171,172]. Curcumin has been shown to inhibit cancer cell metastasis development. It prevents cancer cells from destroying healthy tissue by inhibiting the action of matrix metalloproteinases, which control the mechanism. Curcumin inhibits the expression of genes participating during tumor development, apoptosis, and proliferation, such as c-myc, cyclin D1, Bcl-xL, and Bcl-2. NF-κB suppression is essential in proliferation and carcinogenesis. Curcumin inhibits the function of NF-κB, which may enhance the expression of genes involved in invasion (for example, matrix metalloproteinases), proliferation (for example, c-myc and cyclin D1), and antiapoptosis [173].

Basniwal et al. investigated the antitumor characteristics of curcumin nanostructures in cancer cell lines from the skin (A431), liver (HepG2), and lungs (A549) [174]. In aqueous environments, curcumin nanomaterials have been shown to have a much better impact on cancer cells than natural curcumin. Another study found that PLGA–curcumin nanomaterials increased apoptosis, lysosomal function, suppression of nuclear β-catenin action, and androgen receptor in prostate cancer cells as a consequence of a development blockade [175]. One of the more common genomic subgroups of breast cancer with a metastatic form is triple-negative breast cancer. Exogenous p53 and dendrosomal nanocurcumin have been shown to have antitumor activity on triple-negative breast cancer cells, especially when combined [176]. NF-κB and HIF-1 are both needed for the control of cancer cell development. In the hypoxic microenvironment, PLGA nanostructures packed with curcumin increased the expression of NF-κB and HIF-1 subunits (nuclear p65 (Rel A) and HIF-1α) throughout the lung and breast cancerous cells [177,178].

In cancer therapy, providing appropriate concentrations of therapeutic drugs around the tumor region is essential to eradicate the malignant cells while causing the least amount of harm to normal cells [179,180]. In our opinion, the development of curcumin nanoformulations displaying better anticancer properties is a critical step forward. Different types of nanocarriers have been employed to obtain better characteristics for potential application in cancer therapy [181]. In the case of cancer therapies, the type of preparation determines the durability, effectiveness, and selectivity of a formulation. Researchers believe that because of their biocompatibility, liposomal, poly(caprolactone) (PCL) or poly(lactide-co-glycolide) (PLGA) nanomaterials, and self-assembly curcumin formulations should be given the greatest priority in cancer therapeutic strategies [182,183].

The majority of tabulated formulations (Table 1) report the passive targeting mechanism of curcumin nanostructures rather than the active targeting mechanism. The main aspect of curcumin nanomaterials is that they are passively targeted, and it is this trait that encourages their accumulation in tumors. Several significant characteristics, including particle diameter, zeta potential, and the solubility or dispersion of nanostructures, may influence the effectiveness of passive targeting techniques. All that is observed in nanoformulations of ideal size is the enhanced permeability and retention (EPR) effect, which in turn causes a rise in the concentrations of accumulation in tumors. Additionally, a hydrophilic coating using poly(ethylene glycol) inhibits the interaction between proteins and cells, resulting in a reduction in the opsonization response.

## 4. Curcumin and Nanocurcumins as Drugs: Comparable Properties and Effectiveness

Nanocurcumin’s features are determined not only by their chemical structure but also by their physical characteristics. The particle size, surface charge, hydrophobicity, and surface area of nanocurcumin are essential physicochemical features that allow it to be more potent than traditional curcumin [195]. Previous research has shown that these characteristics, which include strong pharmacological properties and successful target specificity, can contribute to a higher dissolution rate and higher bioavailability via the oral route. Curcumin’s properties change when its particle diameter changes on the nanometer scale. It was discovered that reducing particulate diameter increased the potency of nanocurcumin and elevated it over natural curcumin. Due to its greater surface region, nanocurcumin is deemed an appropriate adjuvant for application as a medicine compared to regular curcumin. Nanocurcumin can penetrate organ systems that curcumin is unable to penetrate [196]. Nanocurcumin was discovered to provide a better intracellular absorption potential than standard curcumin [197,198]. This ability is also essential for detecting intracellular pathogens of infectious disorders. Unlike free curcumin, nanocurcumin has a higher physiological dissolution rate throughout tissues and plasma. In an in vivo study on rats, Ma et al. showed that nanocurcumin improves nanoparticle distribution in tissues and biocompatibility by providing a 60-fold improvement in biological half-life as compared to the natural curcumin treatment [199]. For cerebral malaria trials, Dende et al. discovered that nanocurcumin has a higher bioavailability than conventional curcumin and can prevent degenerative shifts [200]. When a 5 mg oral dosage of a system containing PLGA and 350 μg of curcumin was administered, the level of curcumin within brain tissues increased threefold over that observed with 5 milligrams of pure curcumin. Additionally, it was discovered that curcumin-based nanoformulations improve curcumin’s persistence, circulation, and average residence time within the body [201]. Materials composed of nanostructures provide a greater available surface region, which enhances the aqueous solubility and degradation rate, resulting in increased absorption and bioavailability of the medication. Additionally, a large surface area increases a medication’s reaction to a particular chemical target and its pharmacokinetic properties [202]. Because of the greater surface region, the substance loaded into nanomaterials is released to the particle surface, promoting the rapid release of the drug. Furthermore, the large surface area distinguishes nanomaterials as unique and viable candidates for several applications.

The importance of electrical charges on the surface of curcumin nanomaterials has been identified. Muller and Keck discovered that both positive and negative zeta potential inhibits aggregation of the nanomaterials [203]. As a result, nanomaterials under suspension are highly stable/durable. Due to curcumin’s high poor water solubility, curcumin formulations aggregate and become vulnerable to opsonization, while nanocurcumin, by contrast, decomposes entirely throughout aqueous solutions and forms no clusters, owing to zeta potential inclusion. Compared to negatively charged particulates, a positive charge collected upon the nanostructure surface is often considered ideal. This promotes deep penetration inside the cellular membrane and also strong absorption capacity. Furthermore, nanostructures with a moderate positive charge increase internalization potential, while a high positive charge causes cell toxicity [175]. On the other hand, a negative charge does not promote penetration into the cellular membrane at all, but rather prohibits it from decomposing in such circumstances, and encourages particulate equilibrium throughout the bloodstream. No et al. discovered a connection between the electrical charges of the surface of nanocurcumin and its antibacterial properties. Positively charged curcumin nanostructures demonstrated superior antibacterial properties in *Listeria monocytogenes* [204].

## 5. Cancer Immunotherapy Using Nanocurcumin Formulations

The components of the body’s immune system have a vital function in fighting various cancers. Despite the immune system’s concerted attempts to eliminate them, tumor cells cleverly evade detection by employing a variety of evasion strategies to get around the monitoring system [205]. Current techniques, such as those aimed at blocking immune checkpoint modulators, eliminating immunological tolerance, incorporating modified T cell treatments, and determining new tumor antigens by next-generation sequencing, have offered innovative possibilities for cancer immunotherapies [206]. Cancer immunotherapy can be either passive or active, depending on the kind of cancer [207]. The former involves the administration of substances, such as cytokines, monoclonal antibodies (mAbs), or lymphocytes, that stimulate anti-tumor responses, whereas the latter involves the activation of the immune system to attack malignant cells through nonspecific immunomodulation, vaccination, or targeting specific antigen receptors [208,209]. The transformation of the immunogenicity of the tumor microenvironment into tolerogenicity is regarded as the most important factor in tumor immunity resistance [210]. Different immunotherapies have been developed to inhibit such immunological escape pathways, resulting in improved clinical outcomes. However, there are several downsides to these therapies [211]. Curcumin has been investigated as a possible anti-cancer chemical for some time now. Against the traditional hallmarks of cancer, such as apoptosis evasion, angiogenesis, continuous proliferation, tissue invasion, continuous insensitivity to growth inhibitors, and metastasis, its potentialities have been discovered to be significantly effective [212]. As the diversity of curcumin’s activity has thus already been proven, the investigation of its use with immunotherapies may open a new channel for researchers and clinicians.

Zhu et al. created innovative T7 peptide-modified nanomaterials (T7-CMCSBAPE, CBT) based on carboxymethyl chitosan (CMCS) to address the limitations of free pharmaceuticals and improve therapeutic results. These nanomaterials are capable of selective attachment to the transferrin receptor (TfR) expressed on lung cancer cells, as well as accurately modulating drug release based on the pH levels and the quantity of reactive oxygen species. Docetaxel (DTX) and curcumin exhibited drug-loading concentrations of around 7.82% and 6.48%, respectively. Even at concentrations as high as 500 μg/mL, acceptable biosafety could be achieved. More notably, the T7-CMCS-BAPE-DTX/curcumin (CBT-DC) complexes demonstrated improved anti-tumor activities in vitro and in vivo when compared to DTX monotherapy and other nanostructures packed with DTX and curcumin alone. Moreover, it was shown that CBT-DC could improve the immunosuppressive milieu, hence promoting the suppression of tumor development and growth inhibition [213]. These findings contribute to laying the groundwork for combinatorial lung cancer therapy. Camptothecin (CPT) is a potent anti-cancer drug with a broad spectrum of activity that is effective against various kinds of cancer [214]. Nevertheless, its applicability in glioma therapy has been hampered by the tumor’s immunosuppressive environment and the difficulties associated with medication administration across the blood–brain barrier (BBB). In a study, neurotransmitter analog-modified liposomes (NT-LIP) were synthesized by doping lipidized tryptamine (Tryp) with curcumin and CPT to improve co-delivery and chemo-immunotherapy in glioma patients. Tryp inclusion improves the effectiveness with which CPT and curcumin are delivered throughout the BBB. CPT also inhibits cell proliferation followed by NT-LIP uptake by the cell. A combination of CPT and curcumin reduces the increased expression of the programmed cell death 1 ligand 1 (PD-L1) induced by CPT, thereby preventing the inactivation of T-cells and synergistically increasing the effectiveness of chemo-immunotherapy. Moreover, both Tryp and curcumin prevent the indoleamine 2,3-dioxygenase (IDO) route, which has been shown to decrease regulatory T cell (Treg)-mediated immunosuppression. These compounds have a potency to be used in combination with PD-L1 inhibitory activity to produce synergistic antitumor immunity against tumor cells. Overall, this nanoplatform helps with the delivery of targeted therapies as well as the weakening of the immunosuppressive environment in glioma therapy [215].

Although curcumin has an impact on T cells, it does more than just raise the number of effector T cells or trigger the infiltration of Treg cells; it also promotes the cell-destroying capacity of effector T cells that have been suppressed by tumor cells [216,217]. When tested in vitro, a new nanocurcumin greatly boosted the expression levels of the co-stimulatory molecule CD86 on the surface of dendritic cells, and considerably lowered the concentrations of pro-inflammatory substances released by effector T cells. Curcumin has also been found to be a boost to effector-T-cell-induced cytotoxic effects against esophageal cancer cells [218]. Adoptive T-cell therapy has emerged as a significant area of investigation in the field of tumor immunotherapy. Using autologous reinfusion of effector T cells that had proliferated in vitro, researchers were able to enhance the number of exogenous effector T cells in people while also inducing tumor apoptosis [219]. Nevertheless, because of the large number of immunosuppressive cytokines found in the tumor immunosuppressive microenvironment, the effectiveness of adoptive T-cell treatment is frequently temporary and likely to fail in most cases. Combining adaptive T-cell treatment with curcumin improved the intertumoral cell infiltration of CD8^+^ T cells in the E.G7 mouse T lymphoma model after the course, and curcumin raised levels of interferons (IFN)-produced by CD8^+^ T cells [220]. These findings demonstrated that combination therapy was more effective than adaptive T-cell immunotherapy alone when compared to the latter [221].

## 6. Anti-Angiogenic Activity of (Nano)Curcumin for Cancer Treatment

The co-delivery of various medications with complementary anti-tumor processes using nanocarriers is a successful technique for treating cancer [222]. The concomitant use of medications featuring pro-apoptotic and anti-angiogenic properties may be useful in the treatment of human hepatocellular carcinoma (HCC). An amphiphilic poly(amino ester) copolymer was developed by Zhang et al. to manufacture pH-sensitive nanostructures that delivered Dox and curcumin to provide a potent anti-angiogenesis medication [223]. Curcumin and Dox co-loaded nanostructures were synthesized using an optimum drug ratio, and they demonstrated minimal polydispersity, excellent encapsulation capacity, and improved release inside the acidic environment of cancerous cells. Additionally, compared to the usage of free drugs, increased cellular internalization of payloads supplied by curcumin and Dox co-loaded nanomaterials was seen in human liver cancer SMMC 7721 cells as well as in human umbilical-vein endothelial cells (HUVECs). As reported before, these nanostructures triggered a significant death rate by decreasing mitochondrial membrane potential. These nanoparticles also demonstrated significant anti-angiogenic properties in vivo and in vitro, such as suppression of HUVEC proliferation, migration, invasion, and tube formation, all of which were mediated via the VEGF pathway. If the pro-apoptotic medicine Dox and the antiangiogenic compound curcumin are encapsulated in pH-sensitive nanoparticles, a potential method for inhibiting the progression of HCC can be developed that is both efficient and synergistic. It is becoming increasingly popular in mainstream cancer treatment investigation to synthesize metallic nanostructures to boost therapeutic index and medication transport, and this is a promising technique. In a study by Kamble et al., curcumin-capped copper nanomaterials were examined for their potential as inhibitors of in vivo angiogenesis, pro-angiogenic cytokines implicated in promoting tumor angiogenesis, and inhibitors of MDA-MB-231 breast cancer cell proliferation and migration. Using an in vivo model of the chorioallantoic membrane (CAM), the antiangiogenic ability of the compound was investigated. The effectiveness of curcumin-capped copper nanostructures on the growth of a breast cancer cell line was investigated using a 3-(4,5-dimethylthiazol-2-yl)-2,5-diphenyltetrazolium bromide (MTT)-based cytotoxicity test. The wound healing migration test was utilized for the evaluation of the effects of curcumin-capped copper nanomaterials on the migratory capacity of a breast cancer cell line. Pure curcumin was employed as a reference compound. Curcumin-capped copper nanomaterials were unable to display substantial anti-angiogenic or antitumor activity when compared to free curcumin [224]. Efforts are being made to create silver nanostructures using a green technique, with the aim of employing them as nanocarriers for curcumin. The UV–Vis spectrophotometry was used to monitor the green synthesis of the gum-stabilized Ag nanostructures, and the FT-IR was used to investigate the potential interactions between the gum and the Ag nanomaterials. The diameter, zeta potential, size distribution, polydispersity index, shape, medication load-carrying effectiveness, and interactions with excipients of the curcumin-loaded Ag nanoparticles were all evaluated. The generated nano-anticancer formulations were described and evaluated for their anti-cancer possibilities against three different cell lines: FM-55, MM-138, and MCF-7. The Ag nanomaterials served as good nanocarriers for the enhanced quantity of curcumin. When tested against MM-138, FM-55, and MCF-7 cell lines in an in vitro antitumor investigation, Ag nanomaterials, curcumin, and curcumin-loaded Ag nanostructures had IC_50_ values of 166.3, 82.2, and 61.6 μg/mL; 153.2, 107.3, and 77.1 μg/mL; and 144.6, 81.2, and 60.6 μg/mL, respectively. When it came to loading efficiency for curcumin, silver nanoparticles were shown to be quite effective. Additionally, the curcumin-loaded nanomaterials showed excellent antitumor activity against the MM-138, FM-55, and MCF-7 cell lines. In the future, the nano-anticancer drug formulations that have been disclosed may be investigated in in vivo experiments and clinical trials for cancer therapy [225]. Van der Vlies and colleagues synthesized curcumin-loaded polymeric nanostructures, in which curcumin was complexed with phenylboronic acid-containing framboidal nanostructures through the combination of curcumin and nanoparticles in an aqueous solution. Under physiological conditions, the chemical stability of curcumin-loaded polymeric nanoparticles was enhanced, and the curcumin was released in a consistent and sustained way. Furthermore, in chicken chorioallantoic membrane models, they dramatically increased the antiangiogenic as well as antitumor effects of curcumin [226]. 

## 7. Effects of Curcumin and Nanocurcumin toward Bacteria and Viruses Associated with Cancer

Many human bacteria and viruses are currently recognized to be associated with cancer. In general, these microorganisms are involved in the mechanisms that destroy the pathways dedicated to maintaining the integrity of genetic information, and also in the mechanisms that prevent apoptosis in damaged cells and bring about undesired cellular proliferation. These effects reduce the self-repair ability of their hosts’ cells and eventually result in cellular transformation, progression of cancer, and resistance to therapeutics [227,228]. Zella and Gallo concluded that *Mycoplasma* sp. and possibly other bacteria, such as *Mycoplasma fermentans* (known as DnaKs), may contribute to cellular transformation and stop certain pharmaceutical therapies that depend on functional p53 for their anti-cancer activity. Some examples of bacteria, such as *Helicobacter pylori* (associated with human cancers), *Fusobacterium nucleatum* (associated with colorectal cancer), *Chlamydia trachomatis* (associated with cervical cancer), and *Mycoplasmas* sp. (associated with prostate and colorectal cancer), and viruses including human T-cell leukemia virus type-1 (HTLV-1), human papillomavirus (HPV), hepatitis B Virus (HBV), hepatitis C Virus (HCV), Epstein–Barr Virus (EBV) and human gammaherpesvirus 4 (HHV-4), and Merkel cell polyomavirus (MCV) are determined to be linked to cancer(s) [229].

Curcumin has been shown to have antibacterial effects against Gram-positive and Gram-negative bacteria. These include antibiotic-resistant strains that cause human infections. It is also effective at inhibiting bacterial biofilms that are surrounded by a self-produced polymer network [230,231]. Alam et al. investigated the effects of pure curcumin and curcumin-loaded PLGA nanoparticles against gastric cancer and *Helicobacter pylori*. Results showed a significant improvement in anti-*H. pylori* activity with nanocurcumin rather than pure curcumin, with MIC values of 8 and 16 μg/mL, respectively. In the case of cytotoxicity against AGS cells, the IC_50_ values of nanocurcumin and curcumin were 18.78 and 24.20 μM, respectively, for 72 h. Further cell cycle evaluation revealed that the population of cells in the sub-G0 population increased by 24.5% (for curcumin) and 57.8% (for nanocurcumin), which indicated that the apoptotic cell population was increasing [232].

The antiviral potency of curcumin has been documented against enveloped and non-enveloped RNA and DNA viruses, including Zika, HIV, dengue, influenza, chikungunya, hepatitis, respiratory syncytial viruses, arboviruses, herpesviruses, noroviruses, and papillomavirus [233,234]. Curcumin has demonstrated a potent anti-HPV effect in tests conducted in vitro using cancerous cells derived from the oral and cervical cavities, cervical cancer being the fourth most common cancer in women and the fourth leading cause of cancer death [235,236,237]. In addition to suppressing lytic replication of the virus, curcumin was capable of reducing the pathogenesis of Kaposi’s sarcoma-associated herpesvirus (KSHV) [238]. Aside from these preclinical investigations, researchers also conducted a placebo-controlled, double-blind, randomized Phase II clinical study of curcumin and another formulation of the chemical called “Basant” [239]. Another clinical study into the intravaginal use of curcumin-based capsules or vaginal cream at nighttime for four weeks showed exceptional (about 80%) clearance of the virus [240,241]. Recently, scientists have considered curcumin’s application as an anti-HPV medicine. It has been demonstrated that curcumin therapy can prevent transcription of HPV 16 E6/E7 as early as 6 h after treatment. This treatment also restores the expression of tumor suppressor proteins such as Rb, PTPN13, and p53 [242]. It enhances the apoptosis caused by paclitaxel in HPV-positive human cervical cancer cell lines by acting on the following pathways: NF-kB, p53, and caspase 3 [243]. Therefore, curcumin can reduce the HPV oncoproteins; it also restores Rb, p53, and ptpn13 proteins; and it suppresses the tobacco carcinogen benzo[a]pyrene-induced upregulation of HPV E7. Additionally, curcumin inhibits the higher production of miRNA-21 in cervical cancer by inhibiting the binding of AP-1 to the miRNA-21 promoter [244]. In light of the evidence, it appears that curcumin and perhaps curcumin nanoformulations may act as potent anti-HPV agents and promote the efficient treatment of oral or cervical cancer by sensitizing and targeting malignancy and cancer stem cells. On the other hand, hormonal treatment is currently administrated to patients with hormone-sensitive recurring or metastatic gynecologic malignancies, even though the therapeutic results and response rates are inconsistent [237]. 

## 8. Clinical Trials of Curcumin Nanoformulations

So far, several studies have explored the safety, pharmacokinetic properties, and efficiency of curcumin in treating various human disorders. Clinical experiments have revealed some promising outcomes, since curcumin can arrest or even prevent the development of cancerous cells. Around 210 clinical trials on curcumin application have been documented. Of these, 92 clinical studies were completed, while the status of 32 clinical trials is uncertain. The remaining clinical trials are in various stages of recruitment (active or not recruiting), suspension, termination, completion, and withdrawal. Many clinical trials have shown that nanocurcumin is useful in treating ankylosing spondylitis, multiple sclerosis, amyotrophic lateral sclerosis, chronic kidney disease, metabolic syndrome patients, and malignancies [11]. An investigational clinical trial performed by Ahmadi and colleagues demonstrated that nanocurcumin provides effective and safe therapy for people who suffer from amyotrophic lateral sclerosis [245]. According to the findings of another clinical study conducted by Dolati et al., nanocurcumin can restore the incidence as well as the functionality of Treg cells in individuals with multiple sclerosis [246]. 

## 9. Various Patents on Using Curcumin Nanoformulations against Cancer

Several nanoformulation patents on the basis of curcumin, such as curcumin-loaded polymeric nanomaterials [247,248], curcumin cyclodextrin [17], curcumin-containing nano-emulsions with antioxidant properties [249], curcumin encapsulated in chitosan nanoparticles [250], curcumin oil emulsion [251], curcumin-loaded vesicles [252], liposomal curcumin [253], curcumin bound to fibroin polypeptide, acidic sophorolipid encapsulated curcumin, and magnetic nanoparticles loaded with curcumin [254] have been granted. As described under the patent WO2009105278A2, a technique for producing curcumin-encapsulated chitosan nanostructures using an ionotropic gelation process and delivering them to extra-testicular Sertoli cells was developed. Almost all of the curcumin delivered to the Sertoli cells was distributed throughout the lungs [255]. For the formulation under US patent US8535693B2, it was discovered that topical nanomaterials can be used to reduce inflammation, skin conditions, and mucosal illnesses [256]. Curcumin and an emulsifier/nonionic surfactant combination were produced as nanomaterials by sonication, and then the mixture was tested on mice. Results showed that the manufactured curcumin treatment comprised a consistent granular layer as well as a sufficient epidermal thickness [257]. In their patent, Xianwang et al. described the manufacturing technique and implementation of a curcumin–chitosan–stearic-acid graft micelle on cancer cells [258]. *In vivo* tests demonstrated that curcumin–chitosan–stearic-acid graft micelle can block the growth of MCF-7/Adr, MCF-7, and colorectal cancer cells while causing no adverse side effects. Magnetic nanoparticles containing curcumin were shown to trigger apoptosis throughout cancer cells. Results demonstrated that curcumin was more bioavailable in the mouse when compared to pure curcumin in the laboratories. In addition, nanocurcumin was found to inhibit the formation of pancreatic tumors in mice [254]. To release curcumin into the intestinal tract, Bansal et al. designed a novel nanocrystalline solid dispersion formulation. To make the dry curcumin powder, stearic acid (nanocrystalline solid dispersion) was used, and it was discovered that curcumin bioavailability increased by 15 times when compared to pure curcumin [259]. According to the patent, Pattayil and Jayaphraba’s innovation resulted in the development of curcumin-coated USPION (ultra-small superparamagnetic iron oxide nanoparticles) for biological applications. The scientists showed that a simple one-pot process can be used to make biodegradable and durable curcumin [260]. Nanomaterials containing a mitochondrial targeting moiety are disclosed in the patent WO2013123298A1, which was issued in 2013. Dhar and Marrache developed an environmentally friendly polymeric material containing a terminal OH group (PLGA-b-PEG-OH) that allowed the conjugation of triphenylphosphonium (TPP), resulting in the formation of PLGA-b-PEG-TPP. In this investigation, curcumin-encapsulated nanomaterials were synthesized using the nanoprecipitation technique, and their efficacy in treating neurodegenerative disorders was assessed. It was shown that curcumin nanomaterials provided superior neuroprotection against β-amyloid plaques as compared to free curcumin [261]. However, Moreover, a liposome PLGA-based emulsified hybrid curcumin nanoformulation has been produced. In cancer treatment, this formulation increases bioavailability while simultaneously decreasing QT prolongation [262]. The concept detailed in European Patent EP2649623B states that curcumin-loaded magnetic nanostructures are beneficial for a variety of malignant cell lines and tumors [254]. The curcumin nanoformulation dubbed curcumin-QLife^®^ was created by dissolving curcumin in a pre-heated solution comprising Tween 20 and polyethylene glycol (PEG) 200 and then cooling the solution [263]. *In vivo* studies on humans and rats revealed that curcumin-QLife^®^ had a higher bioavailability when compared to other accessible curcuminoid products currently on the market. In 2017, Liu and colleagues patented another application of curcumin nanoformulation, involving a nanodelivery system based on curcumin and materials such as chitosan and phospholipids [264]. Recently, to increase curcumin bioavailability, Sezgin and Bayraktar created nanodelivery systems based on curcumin and piperine-loaded biopolymer, employing electrospray processes and/or coating processes [265].

## 10. Conclusions and Future Considerations

To the best of our knowledge, turmeric has gained in-depth attention due to its potential therapeutic properties. Curcumin has demonstrated good antitumor capabilities, but its intrinsic dissolution rate, greater metabolic activity, and poor pharmacokinetic features prevent it from being developed into an effective cancer treatment agent. A literature survey revealed that nano-encapsulation strategies improved its pharmacological effectiveness. Curcumin nanoformulations have the potential to provide several benefits, including enhanced effectiveness and tumor targeting, lower overall systemic toxic effects, greater compliance, and ease of administration. Most specifically, curcumin nanoformulations based on PLGA, cyclodextrin assembly, and magnetic nanoparticles are particularly well-suited to cancer treatment. Nevertheless, the adage “there is still space for growth” perfectly describes the rate of curcumin progression as a viable pharmaceutical adjuvant. As a result, there are still numerous challenges and issues that must be addressed before nanocurcumin might be regarded as a viable candidate for medical applications in human disease. So far, many curcumin nanoformulations have been produced to improve tissue targeting, cellular absorption, and tissue effectiveness. Each curcumin nanoformulation investigated in this article has the potential to bring into question many of the signaling mechanisms associated with a wide range of human disorders. Moreover, several of these formulations are only at the proof-of-concept level, and investigations are still ongoing at pre-clinical levels, so researchers have many questions about the possible toxicological effects of curcumin nanoformulation on human health. In fact, the therapeutic efficacy of curcumin-loaded nanomaterials can be improved while their toxicity may be decreased in certain circumstances. Curcumin nanoparticles are not tissue-specific; they are delivered in this case to normal tissue around tumors or malignant cells. Thus, more consideration can be devoted to the improvement of tissue-specific delivery methods based on nano drugs. Allergic reactions, DNA injury, excitotoxicity, and neuroinflammation are all examples of undesirable toxicity caused by nanomedicine-based drug delivery systems. Therefore, the biodegradability and biocompatibility of curcumin nanodrugs need to be thoroughly investigated and documented. Further pre-clinical and clinical research is necessary to attain a thorough understanding of mechanism of action of curcumin nanoformulations. Another practical alternative is the improvement of the antibacterial and antivirus ability of curcumin as well as its nanoformulations, which are administrated in cancer therapy. The current knowledge may then be used to develop drug candidates for the treatment of cancer(s) alone or in conjunction with other therapeutic approaches.

## Figures and Tables

**Figure 1 molecules-27-05236-f001:**
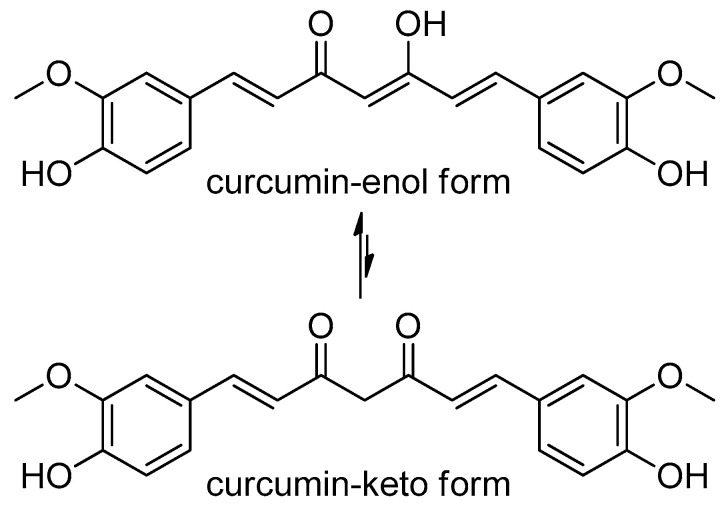
Chemical structure of curcumin in its enol and keto forms.

**Figure 2 molecules-27-05236-f002:**
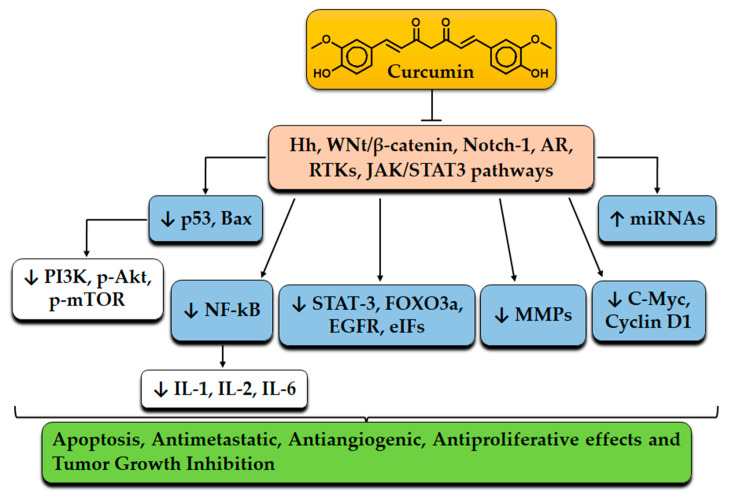
An overview of molecular mechanisms of curcumin against cancer.

**Figure 3 molecules-27-05236-f003:**
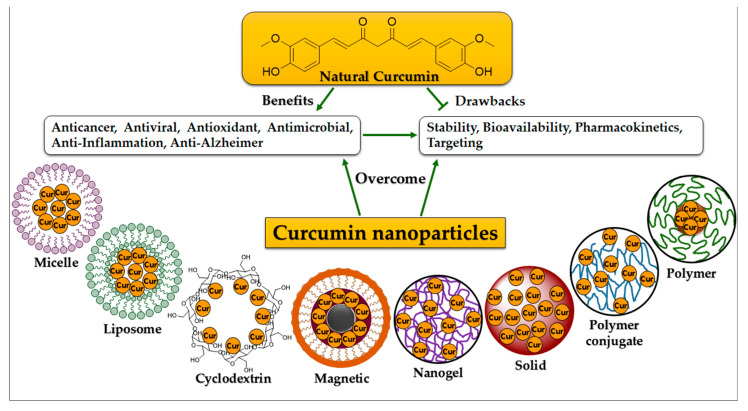
Nano-based formulations of curcumin. Many curcumin-based nanoformulations have a significant impact on pharmaceutical applications, which are effective in the treatment of a wide range of human disorders due to their anti-cancer, antioxidant, antimicrobial, and antiinflammation, and even anti-Alzheimer properties. Most nanoformulations are capable of overcoming curcumin’s weak hydrophobicity, as well as its poor stability and poor cellular bioavailability. Such nanoformulations are utilized for long-term preservation and circulation throughout the body.

**Figure 4 molecules-27-05236-f004:**
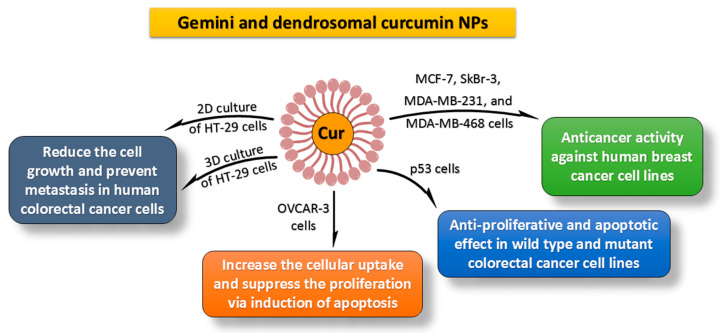
Anti-cancer activity of Gemini and dendrosomal nanocurcumin against various cancer cell lines was examined [155]. Curcumin’s ability to provoke apoptotic cell death, disturb cell cycle activity, and suppress proliferative behavior in cancer cells makes it a promising therapeutic target for human breast, colorectal, lung, carcinoma, prostate, melanoma, myeloma, liver, and pancreatic cancers.

**Table 1 molecules-27-05236-t001:** In vivo and in vitro anticancer potential and mechanism of action of some kinds of curcumin nanoformulations.

Curcumin Nanoforms	In Vitro Cytotoxic Activity	Molecular Mechanism	In Vivo Results	Ref.
Poly(lactide-co-glycolide); PLGA	Cytotoxicity against HCT116, DU145, MDA-MB-231, SEG-1, Jurkat, and KBM-5 cells with IC_50_ < 5 μM.	NF-κB-induced inactivation of and decrease in cyclin D1, MMP-9, and VEGF production.	The half-life of curcumin nanoparticles was 1.75 times longer than curcumin.	[184]
Poly(lactide-co-glycolide); PLGA	Equal cytotoxicity of nanocurcumin and curcumin toward SKBr3, HeLa, and A549 cells.	Increase in Annexin V staining, cleaved PARP expression. Decrease in NF-κB activation.	Not available.	[185]
Poly(lactide-co-glycolide); PLGA	Cytotoxicity against PC-3, LNCaP, and DU145 cells; curcumin-loaded PLGA nanostructures: IC_50_ = 20–22.5 μM; free curcumin: IC_50_ = 32–34 μM.	Inhibition of NF-κB function.	Not available.	[186]
β-cyclodextrin self-assembly of curcumin	In C4-2 and DU145 cells, the curcumin self-assembly concentration was 16.8 μM and 17.6 μM, respectively, which is slightly less than the free curcumin concentration.	Increase in cleaved PARP expression.	Increased curcumin levels in serum concentrations by up to twofold (Unpublished data with Subhash Chauhan Lab)	[17]
MPEG-PCL micelle	Cytotoxicity against C-26 colon cancer cells; Cur-MPEG-PCL micelles: IC_50_ = 5.78 mg·mL^−1^. Free curcumin: IC_50_ = 3.95 mg·mL^−1^.	Not available.	Increase in curcumin concentrations in rat plasma (>2 times) and suppression of subcutaneous C-26 colon cancer development in a xenograft mice model.	[187]
Poly(butyl cyanoacrylate) nanomateriales	Cytotoxicity against Bel7402, HepG2, and Huh7 cells (IC_50_ ≈ 15 μg/mL).	Suppression of VEGF and downregulation of COX-2 expression.	A 2.2-fold reduction in HepG2 tumor volume in a xenograft mice model.	[188]
Dendrosomal curcumin	Cytotoxicity against WEHI-164 cells; IC_50_ = 16.8 & 7.5 μM after 24 & 48 h. Cytotoxicity against A431 cells: (IC_50_ = 19.2 and 14.3 μM after 24 & 48 h.	Increase in cleaved PARP expression and further Annexin V staining (apoptosis).	Reduction in tumor development.	[189]
Self-microemulsifying medication delivery device enhanced with folic acid.	Effective cytotoxicity of folate curcumin-nanoemulsion, curcumin-emulsion, and free curcumin against HeLa cells at concentrations of 18.27, 36.69, and 30.4 μM, respectively. Effective cytotoxicity of folate curcumin-nanoemulsion, curcumin-emulsion, and free curcumin against HT-29 cells at concentrations of 20.57, 38.59, and 25.62 μM, respectively.	Not available.	Increase in folate curcumin-nanoemulsion adsorbsion from 58.41% to 73.38% in 6 h (in situ colon-perfused rats)	[190]
Thermo-sensitive nanocarrier	Showing particular toxic effects on cancer cell lines (KB, MCF-7, and PC-3 cells) while being nontoxic to the L929 cell line.	Increase in apoptosis due to Annexin-A and PI binding.	Not available.	[191]
NanoCurc™	Little or inhibited growth of JHH-GBM14, D283Med, DAOY, and glioblastoma neurosphere lines HSR-GBM1.	G(2)/M arrest and apoptosis induction via the inhibition of STAT3 and Hedgehog signaling pathways.	~0.5% localization of the injected drug within the brain.	[192,193]
Amphiphilic mPEG-palmitic acid polymer	Cytotoxicity against HeLa cells; nanocurcumin; IC_50_ = 15.58 μM, curcumin; IC_50_ = 14.32 μM.	Increasing the anticancer activity in vitro by enzyme-catalyzed release.	Not available.	[194]

## Data Availability

Data sharing is not applicable to this article.

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
