# Peer review of "Curcumin-Based Nanoformulations: A Promising Adjuvant towards Cancer Treatment"

_molecules, 2022, doi:10.3390/molecules27165236_

Round 1

Reviewer 1 Report

Ghoran and co-workers presented a manuscript for Molecules entitled “Curcumin-based Nanoformulations: A Promising Future Opportunity towards Cancer Treatment”. Authors analysed the state of the art of curcumin-based systems for the treatment of cancer diseases. Authors made a deep analysis of the recent literature highlighting the strength of the use of nanomedicine in oncology. It is opinion of this Reviewer that this Review can be accepted after MINOR REVISIONS.

-        Many reported studies present curcumin as an adjuvant in anti-cancer therapy, not as an alternative. More attention should be paid by the authors in the use of this term which can be misleading.

-        Lines 54-56: authors should report more references about this topic. You can consider the following:

Annual Reports in Medicinal Chemistry 2020, 55, 45-75; https://doi.org/10.1016/bs.armc.2020.02.001

Pharmaceutics 202012(1), 6; https://doi.org/10.3390/pharmaceutics12010006

Mar. Drugs 201917(9), 491; https://doi.org/10.3390/md17090491

-        Section 2.2. is very interesting and it should be covered in more detail.

Author Response

Point 1: Many reported studies present curcumin as an adjuvant in anti-cancer therapy, not as an alternative. More attention should be paid by the authors in the use of this term which can be misleading.

Response 1: Thank you so much for your keen eyes, we rechecked the whole manuscript and replaced the term “adjuvant” instead of “alternative”.

Point 2: Lines 54-56: authors should report more references about this topic. You can consider the following:

Annual Reports in Medicinal Chemistry 2020, 55, 45-75; https://doi.org/10.1016/bs.armc.2020.02.001

Pharmaceutics 2020, 12(1), 6; https://doi.org/10.3390/pharmaceutics12010006

Mar. Drugs 2019, 17(9), 491; https://doi.org/10.3390/md17090491

Response 2: The suggested references were considered in the content. They were numbered 5-7 in the reference list.

Point 3: Section 2.2. is very interesting and it should be covered in more detail.

Response 3: We completely agree with the reviewer's suggestion. We covered this section more specifically. Thank you for your good comment.

Reviewer 2 Report

Cancer refers to any one of a large number of diseases characterized by the development of abnormal cells that divide uncontrollably and have the ability to infiltrate and destroy normal body tissue. We can treat it by using natural bioactive compounds and their nanoformulations. Curcumin is one of them. However, the manuscript is well written and organized but, still, needs some modification. After adjusting my comments I consider it for publication in this reputed journal.

1.    Title, abstract, and keywords are precise.

2.   Introduction should be improved and authors also should be mentioned the aim of the study in the last paragraph of introduction precisely.

3.     The major change proposed for the manuscript to have scientific relevance is to make a good methodology following the guidelines of the systematic reviews. If there are no article exclusion criteria, at least a flowchart should be included that shows the organization of the information sought. It is suggested to use the PRISMA method, but any method can be used for this purpose.

4.      Conclusion section should be more specific.

5.      The paper contains some grammatical mistakes and syntax errors.

6.      The authors may discuss and cite the following references.

doi: 10.3390/molecules27072165

doi: 10.3390/cancers14030759

doi: 10.3390/molecules26237109

Author Response

Point 1: Title, abstract, and keywords are precise.

Response 1: Thank you very much for the positive feedback.

Point 2: Introduction should be improved and authors also should be mentioned the aim of the study in the last paragraph of introduction precisely.

Response 2: We revisited the introduction part and improve it accordingly. In the last paragraph of the introduction, the aim of the review was added. Thank you.

Point 3: The major change proposed for the manuscript to have scientific relevance is to make a good methodology following the guidelines of the systematic reviews. If there are no article exclusion criteria, at least a flowchart should be included that shows the organization of the information sought. It is suggested to use the PRISMA method, but any method can be used for this purpose.

Response 3: We have performed a comprehensive literature search from the following databases: ACS publication. Elsevier, Taylor and Francis, Wiley Online Library, MDPI, Springer, and Thieme. However, we have no claim on the systematic review. The following keywords were used to search for the present review: “Nanocurcumin”, “Cancer treatment”, “Nanocarriers”, “Drug delivery”, “Curcumin nanoformulations”, “Cancer immunotherapy”, “Anti-angiogenesis”, “Clinical trials”, “Curcumin patent”, “Mechanism of action”.

We have published several review papers in the same manner. The following are examples of our reviews: DOI: 10.1016/j.micromeso.2021.110967, DOI: 10.3390/md19080410, DOI: 10.3390/molecules26061754, DOI: 10.34172/apb.2023.003, DOI: 10.3390/molecules27031128, DOI: 10.3390/md20080474

Point 4: Conclusion section should be more specific.

Response 4: We have tried to carefully revisit the conclusion part. It was briefed and is well prepared now.

Point 5: The paper contains some grammatical mistakes and syntax errors.

Response 5: We have tried to recheck the whole manuscript and resolve the grammatical mistakes and typos. The English language of the paper was also improved.

Point 6: The authors may discuss and cite the following references.

doi: 10.3390/molecules27072165, doi: 10.3390/cancers14030759, doi: 10.3390/molecules26237109

Response 6: The proposed references were added to the text and highlighted in the reference list.

Reviewer 3 Report

After a critical review of the manuscript, I recommend you the following suggestions.

Please change the sentences.

Suggestions:

1. Line 213- 215 “Liposomal medicines, on the other hand, circulate mostly in the spleen, liver, bone marrow, lung, and other organs and tissues, which contributes to a higher medication therapeutic effect and fewer adverse effects. (Rephrase and replace it with the following line)”

Liposomal formulations have extended plasma circulation to reach the spleen and other organs and tissues to provide a major therapeutic effect and limit non-specific toxicity. 

2. Line 243- 245 “Because of their low cost, ease of production, and versatility, γ-CD, β-CD, its analogs or derivatives were commonly employed to deliver medicines.”

Replace medicines with “Pharmaceutical active agents”.

3. Subheadings (3.25 and 3.26) nanostructures and nanospheres are almost the same. nanospheres come under nanostructures. nanosphere can be shifted to 3.25 from microcapsule 3.26 and aligned with nanostructures.

4. Microcapsule (3.26): There is no study that found microcapsule-loaded curcumin in the manuscript.   you should put some studies describing curcumin-loaded microcapsules for potential targets to cover up the topic.

5. line 175-177: A micelle is a spherical vesicle made up of surfactant molecules with an amphiphilic property that assembles spontaneously in the water [56,57]. It is commonly chosen to transmit medicines (medicines should be replaced with (loaded drugs)

6. Lines 177-178: Polymeric micelles encapsulated by curcumin are better to replace with (curcumin-loaded polymeric micelles) or (curcumin encapsulated polymeric micelles)

7. Lines 190- 191: kinetics linked to release and uptake is better to replace with (release kinetics and cellular uptake).

8. Line 224: which inhibited C-26 murine colon cancer cell proliferation that exhibited (that exhibited is replaced with “and exhibited”)

 9.  Section 3.2.3. Cyclodextrin structures:  line 239. binding medicines should be replaced with binding “drugs”. If you describe a carrier system, better to write loaded drugs rather than loaded medicines.

Please correct it all over the manuscript.

3. Subheadings (3.25 and 3.26) nanostructures and nanospheres are almost the same. nanospheres come under nanostructures. nanosphere can be shifted to 3.25 from microcapsule 3.26 and aligned with nanostructures.

4. Microcapsule (3.26): There is no study that found microcapsule-loaded curcumin in the manuscript.   you should put some studies describing curcumin-loaded microcapsules for potential targets to cover up the topic.

 5. line 175-177: A micelle is a spherical vesicle made up of surfactant molecules with an amphiphilic property that assembles spontaneously in the water [56,57]. It is commonly chosen to transmit medicines (medicines should be replaced with (loaded drugs)

6. Lines 177-178: Polymeric micelles encapsulated by curcumin are better to replace with (curcumin-loaded polymeric micelles) or (curcumin encapsulated polymeric micelles)

7. Lines 190- 191: kinetics linked to release and uptake is better to replace with (release kinetics and cellular uptake).

8. Line 224: which inhibited C-26 murine colon cancer cell proliferation that exhibited (that exhibited is replaced with “and exhibited”)

9.  Section 3.2.3. Cyclodextrin structures:  line 239. binding medicines should be replaced with binding “drugs”. If you describe a carrier system, better to write loaded drugs rather than loaded medicines.

Please correct it all over the manuscript.

Author Response

Point 1: Line 213-215 “Liposomal medicines, on the other hand, circulate mostly in the spleen, liver, bone marrow, lung, and other organs and tissues, which contributes to a higher medication therapeutic effect and fewer adverse effects. (Rephrase and replace it with the following line)”

Liposomal formulations have extended plasma circulation to reach the spleen and other organs and tissues to provide a major therapeutic effect and limit non-specific toxicity. 

Response 1: The sentence “Liposomal formulations have extended plasma circulation to reach the spleen and other organs and tissues to provide a major therapeutic effect and limit non-specific toxicity” was replaced according to the nice suggestion.

Point 2: Line 243-245 “Because of their low cost, ease of production, and versatility, γ-CD, β-CD, its analogs or derivatives were commonly employed to deliver medicines.”

Replace medicines with “Pharmaceutical active agents”.

Response 2: The replacement of “Pharmaceutical active agents” was done.

Point 3: Subheadings (3.2.5 and 3.2.6) nanostructures and nanospheres are almost the same. nanospheres come under nanostructures. nanosphere can be shifted to 3.2.5 from microcapsule 3.2.6 and aligned with nanostructures.

Response 3: As we know that nanostructures and nanospheres are somewhat the same. Your good suggestion worked. The term "nanosphere structures" was moved to the subheading Nano-structures. Therefore, the information on nanospheres was mentioned in the following nanostructures. Thank you.

Point 4: Microcapsule (3.2.6): There is no study that found microcapsule-loaded curcumin in the manuscript. You should put some studies describing curcumin-loaded microcapsules for potential targets to cover up the topic.

Response 4: According to our re-evaluation of literature, there was no microcapsule-loaded curcumin investigation to cover the topic. Therefore, we decided to remove the word “microcapsule structures” from the subheading 3.2.6.

Point 5: Line 175-177: A micelle is a spherical vesicle made up of surfactant molecules with an amphiphilic property that assembles spontaneously in the water [56,57]. It is commonly chosen to transmit medicines (medicines should be replaced with (loaded drugs)

Response 5: The replacement of “loaded drugs” was done.

Point 6: Lines 177-178: Polymeric micelles encapsulated by curcumin are better to replace with (curcumin-loaded polymeric micelles) or (curcumin encapsulated polymeric micelles)

Response 6: The replacement of “Curcumin encapsulated polymeric micelles” was done.

Point 7: Lines 190- 191: kinetics linked to release and uptake is better to replace with (release kinetics and cellular uptake).

Response 7: The replacement of “release kinetics and cellular uptake” was done.

Point 8: Line 224: which inhibited C-26 murine colon cancer cell proliferation that exhibited (that exhibited is replaced with “and exhibited”)

Response 8: The replacement of “and exhibited” was done.

Point 9: Section 3.2.3. Cyclodextrin structures:  line 239. binding medicines should be replaced with binding “drugs”. If you describe a carrier system, better to write loaded drugs rather than loaded medicines.

Response 9: The replacement of “drugs” was done.

Reviewer 4 Report

This review paper could be further considered after major revision.  I have two suggestions regarding to this paper:

1, Please give a summary for  the molecular mechanisms of curcumin against cancer and draw a figure. 

2, Curcumin has also the potential antimicrobial activities , including antibacterial  and antivirus activities (PMID: 35326110).  We know current evidences bacteria and virus also important for the development and progression of cancer.  So please add a section about this point. 

Author Response

Point 1: Please give a summary for the molecular mechanisms of curcumin against cancer and draw a figure. 

Response 1: As per the reviewer's suggestion, a summary of the molecular mechanisms of curcumin against cancer and the related figure was added to the text.

Point 2: Curcumin has also the potential antimicrobial activities, including antibacterial and antivirus activities (PMID: 35326110).  We know current evidences bacteria and virus also important for the development and progression of cancer.  So please add a section about this point. 

Response 2: By use of the valuable clue from the esteemed reviewer, we find that the section suggested is interesting. Therefore, we followed the suggestion and added an extra section entitled “7. Effects of curcumin and nanocurcumin toward bacteria and viruses associated with cancer.”

Round 2

Reviewer 4 Report

This paper in the current state could be accepted.